# SARS-CoV-2 Genomic Variants and Their Relationship with the Expressional and Genomic Profile of Angiotensin-Converting Enzyme 2 (ACE2) and Transmembrane Serine Protease 2 (TMPRSS2)

**DOI:** 10.3390/microorganisms12112312

**Published:** 2024-11-14

**Authors:** Henrique Borges da Silva Grisard, Marcos André Schörner, Fernando Hartmann Barazzetti, Julia Kinetz Wachter, Vilmar Benetti Filho, Rafael Emmanuel Godoy Martinez, Christinni Machado Venturi, Gislaine Fongaro, Maria Luiza Bazzo, Glauber Wagner

**Affiliations:** 1Centro de Ciências Biológicas, Universidade Federal de Santa Catarina, Campus Florianópolis, Florianópolis 88040-900, Santa Catarina, Brazil; vilmarbf98@gmail.com (V.B.F.); glauber.wagner@ufsc.br (G.W.); 2Centro de Ciências da Saúde, Universidade Federal de Santa Catarina, Campus Florianópolis, Florianópolis 88040-900, Santa Catarina, Brazil; marcosschorner@gmail.com (M.A.S.); fernandohb55@gmail.com (F.H.B.); kinetzjulia@gmail.com (J.K.W.); rafaelegmartinez@gmail.com (R.E.G.M.); chrisventuri99@gmail.com (C.M.V.); marialuizabazzo@gmail.com (M.L.B.)

**Keywords:** viral infection, host–parasite, genomic variants

## Abstract

Over the past four years, angiotensin-converting enzyme 2 (ACE2) and transmembrane serine protease 2 (TMPRSS2) have been extensively studied, given their important role in SARS-CoV-2 replication; however, most studies have failed to compare their behavior in the face of different SARS-CoV-2 genomic variants. Therefore, this study evaluated the influence of different variants in ACE2/TMPRSS2 expressional and genomic profiles. To achieve this, 160 nasopharyngeal samples, previously detected with SARS-CoV-2 via RT-qPCR (June 2020–July 2022), were quantified for ACE2/TMPRSS2 expression levels, also using RT-qPCR; SARS-CoV-2 genomic variants, along with polymorphisms in the ACE2/TMPRSS2 coding genes, were identified using nanopore sequencing. In order of appearance, the B.1.1.28, Zeta, Gamma, and Omicron variants were identified in this study. The ACE2 levels were higher when B.1.1.28 was present, possibly due to the ACE2/spike binding affinity; the TMPRSS2 levels were also higher in the presence of B.1.1.28, probably attributable to inefficient usage of the TMPRSS2 pathway by the other variants, as well as to the decrease in protease transcription factors when in the presence of Omicron. The rs2285666 (*ACE2*) polymorphism was less frequent when B.1.1.28 was present, which is befitting, since rs2285666 increases ACE2/spike binding affinity. In conclusion, SARS-CoV-2 genomic variants appear to exhibit varying impacts in regards to ACE2/TMPRSS2 genomic and expressional behavior.

## 1. Introduction

### 1.1. SARS-CoV-2

Presenting with diverse pathogenesis, primarily affecting respiratory, digestive, and neurological systems, coronavirus disease 2019 (COVID-19) has become a new daily variable over the past four years [1,2]. Its etiological agent, SARS-CoV-2, is an enveloped virus belonging to the Coronaviridae family, with a single-stranded, positive sense RNA genome that contains information to translate 16 non-structural proteins (nsp), where the ORF1a region codifies for nsp1 to nsp11, and ORF1b for nsp12 to nsp16. Alongside non-structural proteins, the virus genome translates for four structural proteins, i.e., spike (S), envelope (E), membrane (M), and nucleocapsid (N), as well as six accessory proteins [2,3,4]. For SARS-CoV-2 to infect its host, the viral protein spike must bind to its receptor anchored in the host cell membrane, ACE2 (angiotensin-converting enzyme 2), and the cleavage of this viral protein by TMPRSS2 (transmembrane serine protease 2) must occur so that the viral particle can enter the cell and continue its replication cycle [5,6]. Spike cleavage by TMPRSS2 divides the protein into two functional subunits, S1 and S2, where S1 contains the receptor binding domain (RBD), which influences host–virus interaction, including cell tropism and pathogenicity. Meanwhile, within S2 is located the fusion peptide responsible for mediating viral particle fusion with the host cell membrane through conformational modifications [7]. When inside the host cell, SARS-CoV-2 particles shed, exposing their genome for transcription, translation, and packaging into new viruses [8,9].

Even though coronaviruses exhibit a low genomic replication error rate when compared to other viral families, during the pandemic years, a vast spectrum of SARS-CoV-2 genomic variants was observed [10,11]. In the early stages of the pandemic, the evolution of the new coronavirus was limited for a few months, with little-to-no variation; however, a mutation (D614G) that conferred the virus with a 20% replication advantage when compared to that of the wild type (WT) made the B.1 variant highly prevalent in Europe. So, from October 2020, variants started to present high mutational rates, mainly in the spike coding region [12]. In Brazil, different variants of concern (VOC) and variants of interest (VOI) were identified in different pandemic periods: from February 2020 to November of the same year, there was no main prevalent variant, since there were several introduction points of other variants in the country; from December 2020 to August 2021, the most common variants were Gamma (P.1) and Zeta (P.2); from September 2021 to December 2021, the Delta variant (B.1.617.2) replaced the Gamma as the most prevalent variant; from January 2022 onwards, Omicron (B.1.1.529) took the stage as the main genomic variant, not only on a national scale, but at the international level as well [13].

### 1.2. ACE2 and TMPRSS2

Vastly disseminated within the human body, ACE2 is a protein that, besides playing a role in SARS-CoV-2 infection as a spike receptor, belongs to the renin–angiotensin–aldosterone system (RAAS). RAAS is responsible for maintaining hemodynamic equilibrium and blood vessel tonicity via the regulation of blood pressure, sodium and potassium balance, and interstitial volume [8,14,15,16]. Alongside ACE2, TMPRSS2 also presents an important role in SARS-CoV-2 infection; however, besides the gene location and its protein structure, little is known regarding the role that this protein plays in human homeostasis [17]. In addition to the TMPRSS2 pathway, it has been shown that SARS-CoV-2 can use cathepsins for cleaving the spike protein, where viral particles are internalized by endocytosis and the subunits are cleaved by cathepsin L. Still, cathepsins are non-specific proteases, making their activity less efficient and precise than that of TMPRSS2; thus, SARS-CoV-2 tends to prefer using the latter for its replication cycle [18]. However, Omicron has been described as a preferred cathepsin pathway variant, since its spike protein is better cleaved by it [19,20]. Since both proteins are essential for the viral cycle of SARS-CoV-2, during the last few years, they have served as protagonists in research studies aiming to comprehend the relationship between expression levels and the genomic sequences of ACE2 and TMPRSS2 and COVID-19. Rossi et al. [21] demonstrated that during the infection, a reduced ACE2 expression and increased TMPRSS2 levels in the upper respiratory tract were risk factors for respiratory symptoms in COVID-19.

In parallel with gene expression, single nucleotide polymorphisms (SNPs) in the coding regions for both proteins were demonstrated to have the capacity to modify their morphology and functionality. For ACE2, SNPs could interfere in its affinity with spike, such as the rs2285666 (G14934A-NG_012575.3) and rs4646116 (A6324G-NG012575.3) SNPs that besides being located in an intronic region, increase ACE2/spike affinity, thus facilitating viral entry and dissemination [22,23]. For TMPRSS2, rs2070788 (C66056T-NG_047085.2) is associated with reduced expression of the protease, while rs383510 (A49677G-NG_047085.2) demonstrates an association with a greater risk of SARS-CoV-1 infection. [24]. Even with an already robust understanding of the influence of ACE2 and TMPRSS2 on infections via the new coronavirus, when the additional variable of several SARS-CoV-2 variants is added to the equation, a new and complex set of yet-to-be-studied associations arises. Hence, this study strived to establish a relationship between ACE2 and TMPRSS2 expression profiles and genomic variations with SARS-CoV-2 variants to elucidate their possible influence on the complexity of COVID-19.

## 2. Materials and Methods

### 2.1. Clinical Samples

Nasopharyngeal samples were obtained from a previous study approved by the Universidade Federal de Santa Catarina’s Institutional Ethics Review Board (UFSC/CEPSH), entitled “Avaliação da expressão gênica para Enzima Conversora de Angiotensina 2 (ACE2) e Serina Protease da Proteína Transmembrana 2 (TMPRSS2) em pacientes antes e após infecção por SARS-CoV-2”, under the reference number CAAE: 57722022.2.0000.0121. Sampling occurred from 2020 to 2022, all samples were stored at −80 °C at the Laboratório de Biologia Molecular, Microbiologia e Sorologia (LBMMS/UFSC), and the ones considered for this study met the following selection criteria: a reverse transcription real-time polymerase chain reaction (RT-qPCR) detection cycle (Cq) lower than 25 for SARS-CoV-2 detection, with a sampling date within the periods shown in Figure 1.

Among the samples that followed the selection criteria, 46 samples were randomly selected from Period I (of 109 available), 44 from Period II (of 110 available), 5 from Period III (of 15 available), 21 from Period IV (of 202 available), 25 from Period V (of 190 available), and 19 from Period VI (of 54 available), adding up to 160 nasopharyngeal samples. In Periods IV and V, a lower percentage of samples was selected due to the lower quality of isolated nucleic acids, possibly caused by using phosphate-buffered saline (PBS) instead of Universal Transport Medium (UTM^®^) in these periods for storing samples, hence limiting the number of samples that could be used. Patient data were obtained at sampling, upon presentation of a consent form obtained from the above-mentioned study.

### 2.2. Isolation of Nucleic Acids

Obtention of viral RNA for SARS-CoV-2 genomic analysis was carried out using a QIAmp Viral RNA Mini Kit (Cat. No. 52906, Qiagen, Hilden, Mettmann, Germany^®^), following the manufacturer’s protocol, while using 140 µL of the nasopharyngeal sample. Total RNA, used for gene expression analysis, was isolated following a protocol established by Amirouche et al. [25], which used TRIzol^TM^ Reagent (Cat. No. 15596018, Invitrogen, Waltham, Massachusetts, USA^®^) with 250 µL of the nasopharyngeal sample. DNA extraction for verifying the partial genomic sequences of the ACE2 and TMPRSS2 genes was executed using the ReliaPrep^TM^ Blood gDNA Miniprep System (Cat. No. A5082, Promega, Madison, WI, USA^®^), following the manufacturer’s instructions, while using 200 µL of the nasopharyngeal sample.

### 2.3. RT-qPCR and Absolute Quantification

Once isolated, total RNA was quantified via spectrophotometry using NanoVue Plus (General Electronics, Boston, MA, USA^®^). The resulting value was used to dilute the samples to a value of 5 ng/μL in nuclease-free water, aiming to equalize the parameters for gene expression analysis. Using the diluted samples, the quantification of ACE2 and TMPRSS2 gene expression was carried out using RT-qPCR, along with B2M quantification, the latter being used as an internal control to validate the reaction. For amplification, together with GoTaq^®^ Probe 1-Step RT-qPCR (Cat. No. A6121, Promega, Madison, Wisconsin, USA^®^), specific primers and probes, i.e., PrimePCR^TM^ Probe Assay: ACE2, Human (Cat. No. qHsaCEP0051563, BioRad, Hercules, CA, USA^®^), TMPRSS2, Human (Cat. No. qHsaCIP0028919, BioRad, Hercules CA, USA^®^), B2M, Human (Cat. No. qHsaCIP0029872, BioRad, Hercules CA, USA^®^) were used, following the manufacturer’s protocol. Seeking an absolute quantification of the targets, in addition to the samples, commercially fabricated plasmids for ACE2, TMPRSS, and B2M, with previously known concentrations, were also amplified. These plasmids were designed by Grisard et al. [26].

### 2.4. PCR of SNPs Research Regions

Incrementing the initial quantity of DNA within the samples, polymerase chain reactions (PCR) were carried out to amplify the regions of the ACE2 and TMPRSS2 genes in which the main SNPs exhibiting clinical influence can occur. This reaction used the Phusion Plus DNA Polymerase (Cat. No. F630XL, Thermo Scientific, Waltham, MA, USA^®^), following the manufacturer’s instructions, together with the primers listed in Table 1. The PCR products were visualized in ImageQuantTMLAS 500 (General Electronic, Boston, MA, USA^®^) transilluminator via electrophoresis. Samples that presented amplification of all four regions of interest were further sequenced.

### 2.5. Genomic Sequencing

For obtaining the complete SARS-CoV-2 genome present in the samples, the Midnight RT-PCR Expansion Kit (Cat. No. EXP-MRT001, Oxford Nanopore Technologies, Oxford, Oxfordshire, UK^®^) was used to amplify the regions of the whole viral genome, aiming to enrich the samples for sequencing using the Rapid Barcoding Kit 96 (Cat. No. SQK-RBK114.96, Oxford Nanopore Technologies, Oxford, Oxfordshire, UK^®^), alongside the MinION Mk1B sequencer (Oxford Nanopore Technologies, Oxford, Oxfordshire, UK^®^), following the manufacturer’s protocol. Meanwhile, the PCR products from the SNP research regions were also sequenced using the Rapid Barcoding Kit 96 (Cat. No. SQK-RBK114.96, Oxford Nanopore Technologies, Oxford, Oxfordshire, UK^®^), alongside the MinION Mk1B sequencer (Oxford Nanopore Technologies, Oxford, Oxfordshire, UK^®^), following the manufacturer’s protocol. During both protocols, MinKNOW v23.07.12 software was used to monitor the quantity and quality of reads in real time, as well as to execute the base calling process. Good quality reads were saved in a fastq_pass file and used for further genomic analysis.

SARS-CoV-2 whole genome sequencing fastq_pass files were submitted to barcode removal and read filtering using Guppy v.6.5.7; clean reads were then utilized to assemble the viral genome using the Artic nCoV-2019 v1.1.0 protocol, with a minimum depth of 100× and coverage of 98%. Consensus sequences were used to identify genomic variants using Nextclade v2.14.1. These sequences were made public by submitting them to the GISAID database.

Concurrently, fastq_pass files from the PCR products of the SNPs research regions were also submitted to barcode removal and read cleansing using Guppy v6.5.7. Once trimmed, the reads were then aligned with the reference sequences (ACE2: NG_012575.2/TMPRSS2: NG_047085.2) using minimap2 v2.15-r905, with a minimum depth of 60×; alignment files were then sorted and indexed using samtools v1.18; sorted files were used to assemble the consensus sequence with samtools v1.18, which, in part, were aligned to the reference sequence to verify the existence of SNPs using IGV v2.16.2.

### 2.6. Statistical Analysis

After the employment of a Kolmogorov–Smirnov normality test, an independent samples Kruskal–Wallis test was carried out to compare the ACE2 and TMPRSS2 absolute gene expression values within the samples detected with the different viral variants. To evaluate a possible difference in the occurrence of SNPs in samples detected with distinct viral variants, a Pearson’s chi-squared test was employed. Both tests were executed using IBMM^®^ SPSS^®^ Statistics v.22, and all graphs were created using GraphPad Prism v.9.5.2.

## 3. Results

### 3.1. SARS-CoV-2 Genomic Sequencing

Using pre-established sampling periods, it was possible to verify which SARS-CoV-2 genomic variants were more prevalent among them. When observing Figure 2 in Period I, the B.1.1.28 variant was the most common, while in Periods II and III, the Zeta and Gamma variants were the most prevalent. From Period IV onwards, Omicron became the dominant variant, even though one case of Delta variant was observed in that period.

Regarding the results shown in Figure 2, they agree with the results shown by Padilha et al. [27], who monitored the SARS-CoV-2 variants within the same period in the state of Santa Catarina, the same Brazilian region in which the present study was executed. The Delta variant was not included in the analysis, given that it presented with only one detection.

### 3.2. ACE2 Expression When Infected with Different SARS-CoV-2 Variants

By observing ACE2 expression levels for the different SARS-CoV-2 genomic variants (Figure 3), it was noted that when B.1.1.28 is present, a higher ACE2 expression is observed than that seen for the Zeta, Gamma, and Omicron variants.

### 3.3. TMPRSS2 Expression When Infected with Different SARS-CoV-2 Variants

TMPRSS2 levels were demonstrated to be significantly higher in samples infected with B.1.1.28 when compared to those for Zeta, Gamma, and Omicron (Figure 4). Moreover, samples infected with Omicron presented a higher TMPRSS2 expression when compared to that of Gamma.

### 3.4. ACE2 and TMPRSS2 SNPs When Exposed to SARS-CoV-2 Variants

All samples in the present study were amplified for the four SNP research regions. PCR products of 28 of them can be observed in Figure 5, alongside the 100 base pairs (bp) ladder.

By observing the frequency of the SNPs in the ACE2 and TMPRSS2 genes in the presence of different SARS-CoV-2 variants (Figure 6), it was determined that the rs2285666 SNP occurred significantly less frequently when B.1.1.28 was present. For the other explored SNPs, it was not possible to establish any significant difference; the frequencies are demonstrated in Table 2.

### 3.5. ACE2 and TMPRSS2 Expression in Health Professionals

Given that 72 samples of the present study are from health professionals, an opportunity was identified to evaluate the possible differences between ACE2 and TMPRSS2 expression in health professionals and non-health professionals. There was no significant difference when comparing SNP frequencies between the two groups; however, health professionals presented with a significantly higher ACE2 expression, as can be seen in Figure 7.

## 4. Discussion

It is known that the binding affinity of spike and ACE2 is not uniform among different viral variants, changing given the number of mutations present in the spike coding gene, specifically in the RBD [19,28,29]. With mutations present in the RBD, Gamma and Zeta present a significantly higher ACE2/spike binding affinity when compared to that of the wild type [30,31]. Omicron displays four times the number of mutations than previous variants, exhibiting eight substitutions in the ACE2 binding residues, thus increasing the ACE2/spike binding affinity by three-fold when compared to that of the wild type [19,28]. By analyzing Figure 3, it can be suggested that when SARS-CoV-2 variants that display a higher affinity for ACE2 (Zeta, Gamma, and Omicron) are present, a lower ACE2 expression can be observed. So, a higher binding affinity to ACE2 is identified as a potential factor for the reduction of ACE2 expression, potentially indicating that variants adapted with a stronger binding affinity can complete the viral cycle without the need for a vast availability of ACE2 receptors.

ACE2 expression is shown to be induced by SARS-CoV-2 infection via an increase in the expression of the interferon-stimulated gene factor 3 (ISGF3), which displays the interferon stimulated response element (ISRE) as its receptor. After activation of interferon signaling, as occurs in a SARS-CoV-2 infection, ISGF3 is produced. However, the genomic sequence of ISRE has a homologous similarity with an ACE2 locus, leading to a possible ISGF3 binding to an ACE2 promoter, stimulating its expression [32]. Currently, there is no evidence showing a difference in this stimulation when different SARS-CoV-2 variants are present. Still, by observing the results demonstrated in Figure 3, and considering that this stimulus occurs, regardless of the variant, the hypothesis that a higher binding affinity with ACE2 possibly diminishes its expression is reinforced.

It has been shown that Omicron inefficiently uses the TMPRSS2, pathway since its spike is not properly cleaved by this protease due to the high number of mutations in the spike coding region [19,28]. So, when a reduced TMPRSS2 expression is observed while Omicron is carrying out the infection (Figure 4), the non-utilization of the protease could be a contributing factor to this reduction.

TMPRSS2 expression is also indirectly induced by SARS-CoV-2 infection, where the protein expression is regulated by GATA-binding factor 2 (GATA2), which is parallelly involved in the expressional control of several pro-inflammatory cytokines, such as IL1β [33,34]. In a SARS-CoV-2 infection, GATA2 is activated to increase the expression of IL1β, consequently inducing TMPRSS2 production [35]. With that, it was demonstrated by Korobova et al. [36] that in infections where Omicron is present, IL1β levels are reduced, possibly indicating that GATA2 is being less activated, potentially leading to a reduction in TMPRSS2 levels. Therefore, it appears that in the presence of Omicron, the TMPRSS2 pathway is less used, and a potential reduction in the protease’s transcription factors occurs, which is in agreement with the results shown in Figure 4.

The expression levels of TMPRSS2 were also reduced in samples with the Gamma and Zeta variants when compared to those for B.1.1.28 (Figure 4). Although there is a lack of studies describing a possible relationship between the Gamma and Zeta variant infections and a reduced TMPRSS2 expression, in the present study, it was possible to verify the whole genomic sequences of the variants, and it was observed that after Omicron, Gamma presented with the most mutations in the spike coding region, followed by Zeta. Therefore, given the number of mutations in the spike coding region, a similar behavior to that of Omicron, in which the spike is not well cleaved, may occur for both the Zeta and Gamma variants.

One of the shared mutations between Omicron and Gamma leads to an H655Y substitution that when present, reduces TMPRSS2 expression [37], but as can be observed in Figure 4, when Omicron is present, a higher TMPRSS2 expression is observed when compared to that of Gamma, which slightly contradicts the above-mentioned hypothesis. There is no well-described explanation for this phenomenon; however, by analyzing the whole genomic sequences of the variants, a few mutations present in Gamma’s spike RBD and not present in Omicron’s can be identified, such as K417T, E484K, and N501Y. These mutations could lead to an even lower TMPRSS2 expression, but further research is needed to verify this suggestion.

By observing the frequency of the SNPs in the ACE2 and TMPRSS2 genes when in the presence of different SARS-CoV-2 variants (Figure 6), it is determined that the rs2285666 SNP was expressed significantly less frequently when B.1.1.28 was present. According to Sheikhian et al. [38], when rs2285666 is absent, there is an increase in COVID-19 mortality rates when the infection is caused by variants closer to the wild type, and when Omicron is present, the frequency of rs2285666 is significantly higher. Given that in the period when B.1.1.28 was more prevalent (April–October 2021), there was an elevated number of deaths in Brazil (LACEN-SC, 2021; MS, 2024), and that within the variants found in this study, it is the closest to the wild type genotype, the low frequency observed when B.1.1.28 was present is coherent, as is the high frequency when Omicron carried out the infection, going by the above-mentioned study.

When rs2285666 is present, the affinity of the ACE2/spike is increased, since it alters mRNA splicing [22]. So, it can be noted that in samples with Zeta, Gamma, and Omicron detected, there was a lower ACE2 expression, as well as an increased rs2285666 frequency (Figure 3 and Figure 6), complementing the hypothesis that a higher ACE2/spike binging affinity can lead to a decrease in ACE2 levels. Still, in the only sample in which rs4646116 was identified, Omicron was detected, and as this SNP also increases ACE2/spike affinity, these results also support the above-mentioned hypothesis.

During the pandemic period, health professionals were identified as an important sentinel group for SARS-CoV-2 infection and variant monitoring [39]. As 45% of the samples used in this study are from health professionals, an opportunity was identified to evaluate the possible differences between ACE2 and TMPRSS2 expression in health profesionals and non-health professionals. Despite not having any scientific evidence showing a possible explanation for the difference in ACE2 expression levels shown in Figure 7, the fact that during the pandemic period, the population of health professionals was more likely to be subjected to higher exposure to SARS-CoV-2 makes previous contact with the virus much more probable. As mentioned previously, contact with SARS-CoV-2 stimulates ACE2 expression, possibly explaining the divergence observed in Figure 7. Still, data regarding adherence to mask usage, vaccination, and epidemiological policies were not obtained, but these could be major factors influencing this difference in receptor expression profiles [40,41].

## 5. Conclusions

Even with major scientific research being conducted over the past four years regarding ACE2 and TMPRSS2 and their influence on SARS-CoV-2 infection, the complexity generated by viral genomic variants has created new research gaps in need of exploration. Hence, the present study demonstrated that ACE2 and TMPRSS2 behavior varies according to the SARS-CoV-2 genomic variant carrying out the infection, possibly influencing the complexity of COVID-19. In addition, with health professionals having a higher expression of ACE2 when compared to non-health professionals, a higher exposure to SARS-CoV-2 in previous years shows itself as a potential receptor modulator, possibly affecting the course of the infection.

## Figures and Tables

**Figure 1 microorganisms-12-02312-f001:**
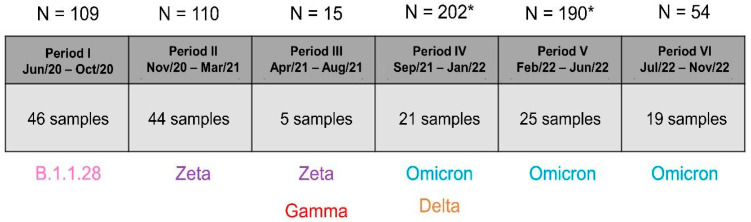
Description of sampling periods, along with the most prevalent SARS-CoV-2 variant, the number of selected samples, and the total number of samples available for each period. Each sampling period (Period I–Period IV) was 4 months long. Below the table are listed the most prevalent SARS-CoV-2 variants within each sampling period, and above the table, the total number of available samples that followed the selection criteria is included. * Indicates a greater difference between the number of selected samples and the total number of available samples.

**Figure 2 microorganisms-12-02312-f002:**
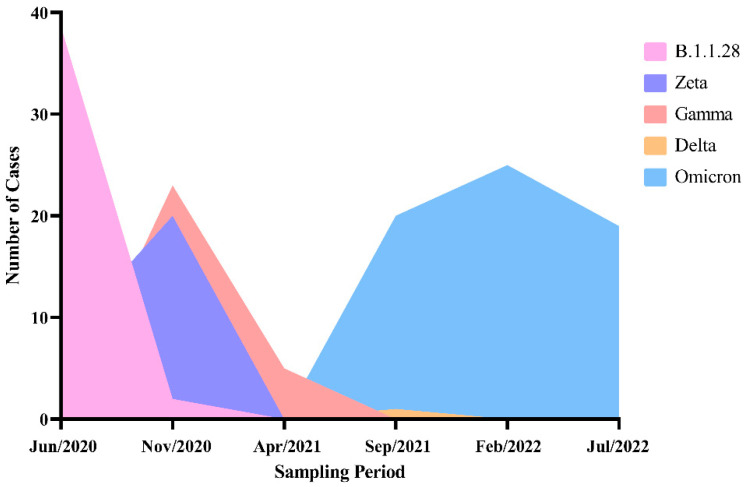
Number of cases of each SARS-CoV-2 genomic variant within the sampling period. Demonstration of the number of identified cases for different SARS-CoV-2 genomic variants during the sampling periods.

**Figure 3 microorganisms-12-02312-f003:**
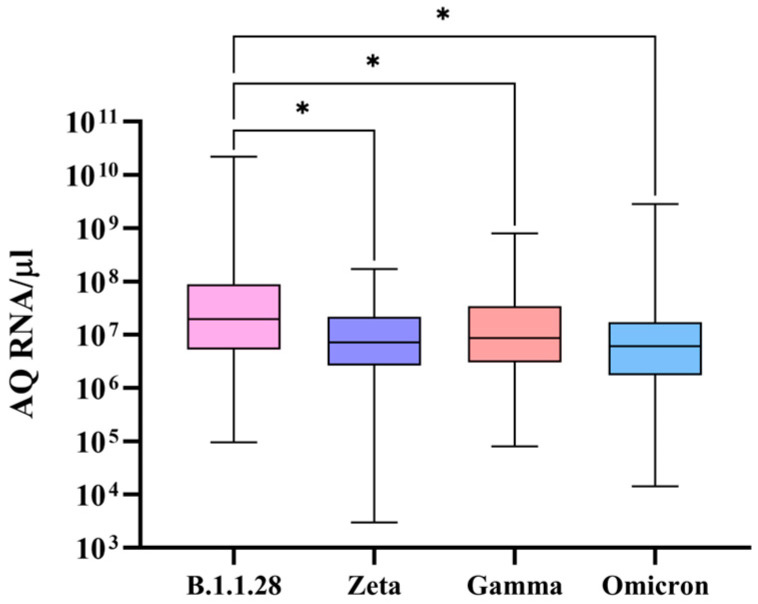
ACE2 expression in nasopharyngeal samples infected with different SARS-CoV-2 genomic variants. Comparison of ACE2 expression levels in nasopharyngeal samples detected for SARS-CoV-2 variants B.1.1.28, Zeta, Gamma, and Omicron. After the employment of an independent samples Kruskal–Wallis test, significant differences were established, represented by * *p* < 0.05. The *Y*-axis plots the absolute quantity (AQ) of RNA per microliter of sample, while the *X*-axis plots the different genomic variants. Box plots display the median (line), interquartile range (box), and minimum and maximum values (bars).

**Figure 4 microorganisms-12-02312-f004:**
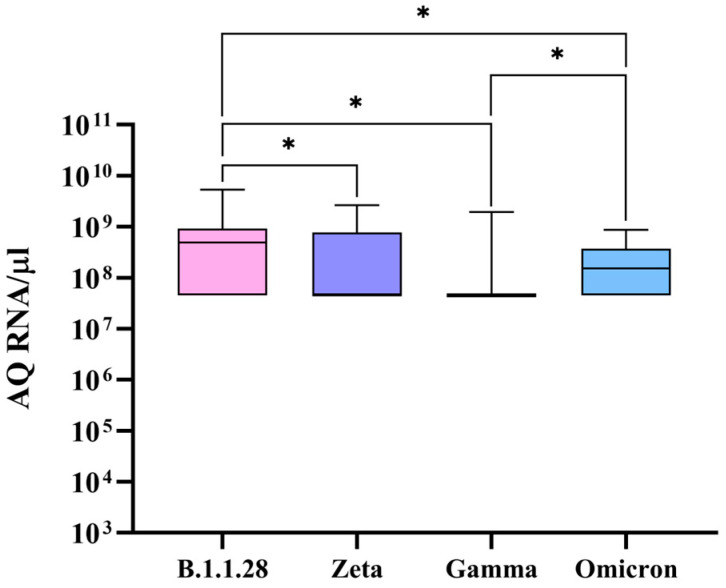
TMPRSS2 expression in nasopharyngeal samples infected with different SARS-CoV-2 genomic variants. Comparison of TMPRSS2 expression levels in nasopharyngeal samples detected with SARS-CoV-2 variants B.1.1.28, Zeta, Gamma, and Omicron. After the employment of an independent samples Kruskal–Wallis test, significant differences were established, represented by * *p* < 0.05. The *Y*-axis plots the absolute quantity (AQ) of RNA per microliter of sample, while the *X*-axis plots the different genomic variants. Box plots display the median (line), interquartile range (box), and minimum and maximum values (bars).

**Figure 5 microorganisms-12-02312-f005:**
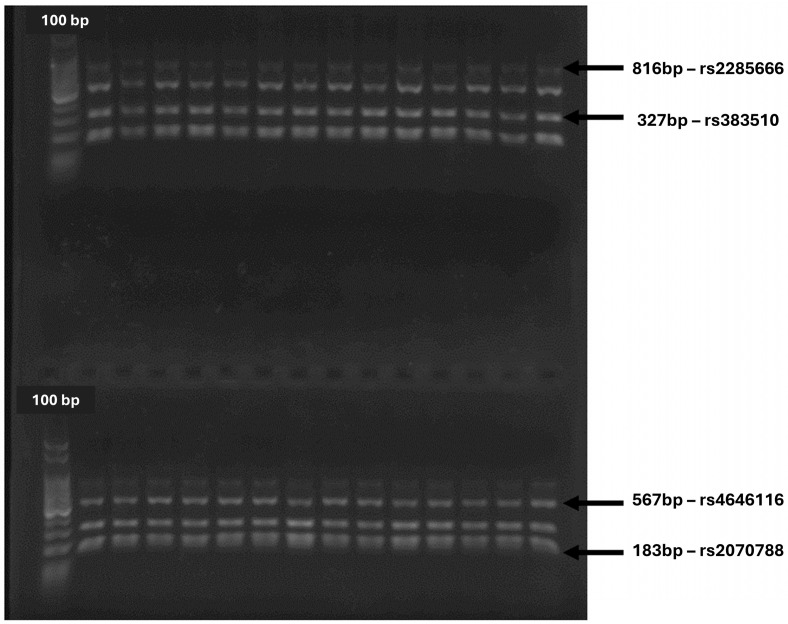
Agarose gel with PCR products for the four SNPs research regions. Picture of the 1.5% agarose gel with PCR products of the multiplex PCR for the four SNPs research regions, alongside a 100 bp ladder; bp: base pairs.

**Figure 6 microorganisms-12-02312-f006:**
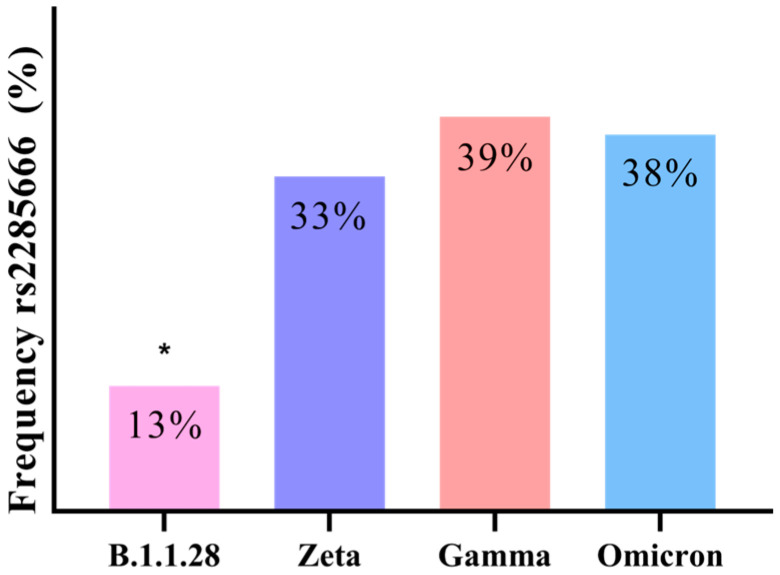
Frequency of rs2285666 in nasopharyngeal samples with the presence of different genomic SARS-CoV-2 variants. Comparison of the frequency of rs2285666 in nasopharyngeal samples detected for SARS-CoV-2 variants B.1.1.28, Zeta, Gamma, and Omicron. After the employment of a Pearson’s chi-squared test, significant differences were established, represented by * *p* < 0.05. The *Y*-axis plots the frequency of rs2285666 SNP, while the *X*-axis plots the different genomic variants.

**Figure 7 microorganisms-12-02312-f007:**
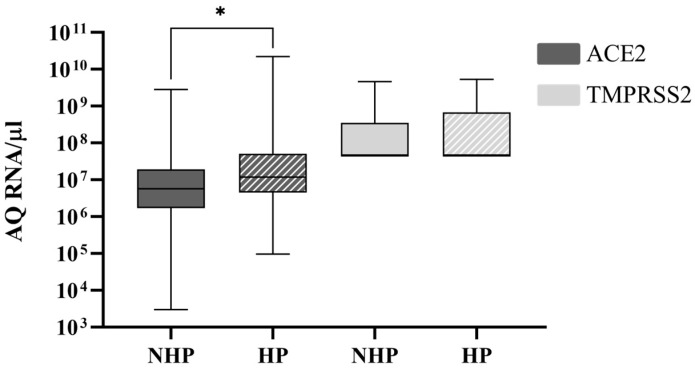
ACE2 and TMPRSS2 expression in health professionals’ and non-health professionals’ nasopharyngeal samples. Comparison of ACE2 and TMPRSS2 expression in SARS-CoV-2-detected nasopharyngeal samples of health professionals (HP) and non-health professionals (NHP). After the employment of an independent samples Kruskal–Wallis test, significant differences were established, represented by * *p* < 0.05. The *Y*-axis plots the absolute quantity (AQ) of RNA per microliter of sample, while the *X*-axis plots the results for the health professionals (HP) and non-health professionals (NHP). Box plots display the median (line), interquartile range (box), and minimum and maximum values (bars).

**Table 1 microorganisms-12-02312-t001:** Primer sequences used to amplify the regions of interest for SNPs in ACE2 and TMPRSS2.

SNP	Forward Primer	Reverse Primer	Target Gene	Reference Article
rs2285666	TTCTCCCTGCTCCTATACTACCG	TTCATTCATGTCCTTGCCCTTA	*ACE2*	[22]
rs4646116	ACCGGTTTTGATTTGGCCAT	CCCTTTTCAGTTTCACGGGC	*ACE2*	[23]
rs2070788	GAAGTGCTTAGTGGCAGGCA	AGTTTCTGCTGATGAGGAGCC	*TMPRSS2*	[24]
rs383510	ATGGCTGTGCTTGGGAAATAAC	CTTATTTCCTGGCCGGACGC	*TMPRSS2*	[24]

**Table 2 microorganisms-12-02312-t002:** Occurrence frequencies of SNPs in ACE2/TMPRSS2 coding genes within the nasopharyngeal samples detected with different SARS-CpV-2 genomic variants.

SNP	B.1.1.28	Zeta	Gamma	Omicron
rs2285666	13%	33%	39%	38%
rs4646116	-	-	-	1.4%
rs2070788	83%	70%	79%	86%
rs383510	83%	89%	82%	81%

## Data Availability

All SARS-CoV-2 whole genome sequences obtained via nanopore sequencing were made publicly available in GISAID database (EPI_SET_241113hb).

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
