# Peer review of "SARS-CoV-2 Genomic Variants and Their Relationship with the Expressional and Genomic Profile of Angiotensin-Converting Enzyme 2 (ACE2) and Transmembrane Serine Protease 2 (TMPRSS2)"

_microorganisms, 2024, doi:10.3390/microorganisms12112312_

Round 1
Reviewer 1 Report
Comments and Suggestions for Authors
Authors investigated the correlation between SARS-CoV2 variants and their receptors- ACE2 and TMPRSS2. The study design is appropriate and the results are well presented. Discussion is comprehensive. Methods can be improved by adding more information on the protocols followed. Conclusions can also be improved significantly. I have few other minor suggestion, as following:
Minor comments:
Line 105: 'previously study' should be 'previous study'.
Methods: Please add the catalogue numbers of kits used.
Line 132-139: More details are needed, such as volume of sample used for extraction etc.
English should be proof-read throughout the manuscript.
Comments on the Quality of English LanguageEnglish can be improved. Please proof read carefully.
Author Response
Comment 1: “Line 105: 'previously study' should be 'previous study'.”
Response 1: Thank you for your correction. It is done.
Comment 2: “Methods: Please add the catalogue numbers of kits used.”
Response 2: Thanks for pointing that out. We included kit catalog numbers in lines 134, 137, 140, 149, 150, 151, 160, 167, 169, and 172.
Comment 3: “Line 132-139: More details are needed, such as volume of sample used for extraction etc.”
Response 3: We appreciate your insight. Sample volumes were included in lines 135, 137, and 140.
Comments on the Quality of English Language: “English can be improved. Please proofread carefully.”
Response: Thank you for your comment. We revised the English in this second version of the manuscript.
Reviewer 2 Report
Comments and Suggestions for Authors
Dear authors,
I have now completed the review of the manuscript titled "SARS-CoV-2 Genomic Variants and their relationship with the expressional and genomic profile of Angiotensin Converting Enzyme 2 (ACE2) and Transmembrane Serine Protease 2 (TMPRSS2)."
In the present study, authors examined an important and timely topic - the relationship between SARS-CoV-2 variants and expression of key host proteins (ACE2 and TMPRSS2) involved in viral entry. The researchers used appropriate molecular biology techniques like RT-qPCR and genomic sequencing to analyze viral variants and host gene expression. The study design included samples from multiple time periods, allowing examination of different predominant variants over the course of the pandemic.
The inclusion of health professionals as a subgroup for analysis is interesting and potentially insightful. The statistical analyses used appear appropriate for the data collected.
However, while this study provides some intriguing preliminary data on the relationship between SARS-CoV-2 variants and host gene expression, the small sample size and observational nature limit the strength of the conclusions. Further research with larger cohorts and more rigorous controls would be needed to validate and extend these findings. Especially:
1. The sample size is relatively small (160 total samples), especially when divided among multiple variants and time periods. This limits the statistical power and generalizability of the findings.
2. The study is observational and cannot establish causal relationships between variants and host gene expression changes.
3. Only nasopharyngeal samples were examined. Expression patterns may differ in other relevant tissues like lung epithelium.
4. The study does not account for potential confounding factors that could influence ACE2/TMPRSS2 expression, such as age, sex, comorbidities, or medications.
5. The functional significance of the observed gene expression changes is not explored. The clinical relevance remains unclear.
6. The discussion speculates quite a bit about potential mechanisms without direct evidence. More caution in interpretation may be warranted.
7. The study lacks a control group of uninfected individuals for comparison of baseline ACE2/TMPRSS2 expression.
8. Only a few select SNPs were examined. A more comprehensive genetic analysis may reveal other relevant polymorphisms.
9. The paper does not adequately address potential batch effects or technical variability in sampling/processing over the extended time period.
10. The conclusions drawn about differences between variants seem somewhat overstated given the limitations of the data.
In conclusion, I would like to recommend reject with resubmission encouraged.
Author Response
Response: Besides your suggestion for rejection, we are thankful for your insights, and your review.

Reviewer 3 Report
Comments and Suggestions for Authors
Abstract:
1) It is not correct to say that ACE2 and TMPRSS2 have a crucial role in the virus replication. Indeed, ACE2 is important for viral entry, but there are many other players with significant role in viral replication.
2) I don't understand the scientific relevance of the sentence "In order of appearance, B.1.1.28, Zeta, Gamma, and Omicron variant were identified in this study. "
3) The use of whilst at the end of line 26 is inappropriate. Also, the whole sentence from lines 25 to 29 is very unclear.
4) The SNPs should be written in relation for the corresponding genes.
Introduction: The article begins with one completely unacceptable statement and absolutely incorrect citation. Every scientist studying the SARS-CoV-2 virus knows that it has four structural proteins - S, E, M and N. It is also well known that the RdRP is the Nsp12, which is non-structural protein, not an accessory one. The authors, on the other hand, state that SARS-CoV-2 has five structural proteins, and they include Hemagglutinin Esterase (HE), which is not part of the SARS-CoV-2 structure and state that Nsp12 is an accessory protein, which they support by two citations - reference 3 and 4. Nowhere in these articles, one can find this information, since it is not correct. Also in this sentence instead of "several", they should write "six accessory proteins".
Material and Methods:
1)Figure 1 should be a table. The legend is unclear and if they change it to table format will be much easier to understand the data in it.
2) In section 2.2 Isolation of nucleic acids there is no need to clarify what TRIzol is. Is a patented solution by Invitrogen and they should use the brand, not clarify its composition.
3) In section 2.4 PCR of SNPs Research Regions Table 1 should be revised and they should put a legend for the Table, not the text from the template (lane 161). In the table, Primer Nucleotide Sequence 5' should be changed to forward primer and Primer Nucleotide Sequence 3' - reverse primer. The SNPs should be written in relation to the corresponding genes.
Results:
1) They should change the scale for Figure 4 since one can't see the minimal value of the box whisker plot.
2) Fig. 5 - it is more correct to write the size of the bends of the ladder.
3) There is a high degree of similarity between the sentence in lines 253-255 in section 3.5 and lines 354-357 in the Discussion section.
4) What is the meaning of health and non-health professionals?
Discussion:
The whole first paragraph should be moved to the Results section, under Figure 2. The Discussion sometimes sounds like duplicated Results. It is not well written and due to the long sentences the reader can lose the logic, when there is one. I think that the whole discussion should be rewritten in a more scientific manner.
Conclusions: When I read the Conclusions section I thought "So what". Same as the Discussion - it should be written in a more scientific manner.
Comments on the Quality of English LanguageThe authors use very long sentences which causes the lost of the whole concept. Few examples - lines 293-298; 324-330; 346-350. The use of meanwhile and nonetheless in the beginning of some paragraphs in the Discussion section is inappropriate. Throughout the hole text there are a lot if word which are used inappropriate.
Author Response
Comment 1: “Abstract - It is not correct to say that ACE2 and TMPRSS2 have a crucial role in the virus replication. Indeed, ACE2 is important for viral entry, but there are many other players with significant role in viral replication.”
Response 1: Thank you for your insight, agreed. Changes were made in lines 18 and 75.
Comment 2: “Abstract - I don't understand the scientific relevance of the sentence "In order of appearance, B.1.1.28, Zeta, Gamma, and Omicron variant were identified in this study.”
Response 2: We appreciate your feedback. Genomic variants are one of the main topics in the manuscript, thus we considered that listing the variants found in the study as an important result to include in the abstract.
Comment 3: “Abstract - The use of whilst at the end of line 26 is inappropriate. Also, the whole sentence from lines 25 to 29 is very unclear.”
Response 3: Thank you for your comment. We removed the incorrectly used “whilst” and made some changes to the sentences from lines 25 to 28 aiming for better clarity.
Comment 4: “Abstract - The SNPs should be written in relation for the corresponding genes.”
Response 4: We appreciate your suggestion. We included the corresponding gene for the SNP mentioned in the abstract in line 29.
Comment 5: “Introduction - The article begins with one completely unacceptable statement and absolutely incorrect citation. Every scientist studying the SARS-CoV-2 virus knows that it has four structural proteins - S, E, M and N. It is also well known that the RdRP is the Nsp12, which is non-structural protein, not an accessory one. The authors, on the other hand, state that SARS-CoV-2 has five structural proteins, and they include Hemagglutinin Esterase (HE), which is not part of the SARS-CoV-2 structure and state that Nsp12 is an accessory protein, which they support by two citations - reference 3 and 4. Nowhere in these articles, one can find this information, since it is not correct. Also, in this sentence instead of "several", they should write "six accessory proteins".”
Response 5: We are thankful for your correction. Agreed, we corrected this statement and included another citation in lines 42, 43, and 44.
Comment 6: “Materials and Methods - Figure 1 should be a table. The legend is unclear and if they change it to table format will be much easier to understand the data in it.”
Response 6: Thank you for your suggestion, but we think that as an image, compared to the same data in a table, it is more didactic and easier to understand.
Comment 7: “Materials and Methods - In section 2.2 Isolation of nucleic acids there is no need to clarify what TRIzol is. Is a patented solution by Invitrogen and they should use the brand, not clarify its composition.”
Response 7: We appreciate your suggestion. Agreed, we made those changes in line 137.
Comment 8: “Materials and Methods - In section 2.4 PCR of SNPs Research Regions Table 1 should be revised and they should put a legend for the Table, not the text from the template (lane 161). In the table, Primer Nucleotide Sequence 5' should be changed to forward primer and Primer Nucleotide Sequence 3' - reverse primer. The SNPs should be written in relation to the corresponding genes.”
Response 8: Thank you for your insight. We agree the proposed changes were made in Table 1.
Comment 9: “Results - They should change the scale for Figure 4 since one can't see the minimal value of the box whisker plot.”
Response 9: Thank you for your comment. Since TMPRSS2 AQ/µl values roamed around the detection limit of our reaction, most values are close to the minimal value. So even if we change the scale, the difference won´t be apparent.
Comment 10: “Results - Fig. 5 - it is more correct to write the size of the bends of the ladder.”
Response 10: We appreciate your suggestion. Given that our ladder bands aren’t too visible, we preferred to indicate the size of the amplicon bands.
Comment 11: “Results - There is a high degree of similarity between the sentence in lines 253-255 in section 3.5 and lines 354-357 in the Discussion section.”
Response 11: Thank you for your insight. Agreed, we removed the sentence from line 254.
Comment 12: “Results - What is the meaning of health and non-health professionals?”
Response 12: We appreciate your question. At the moment of sampling patients filled out a consent form, and one of the questions was if they were health professionals (any of the health-related professions). With that response, we could divide the patients into groups of self-declared health, and non-health professionals.
Comment 13: “Discussion - The whole first paragraph should be moved to the Results section, under Figure 2. The Discussion sometimes sounds like duplicated Results. It is not well written and due to the long sentences, the reader can lose the logic, when there is one. I think that the whole discussion should be rewritten in a more scientific manner.”
Response 13: Thank you for your comment. The first paragraph of the discussion was moved to the suggested location, and modifications were made to the whole discussion.
Comment 14: “Conclusions - When I read the Conclusions section I thought "So what". Same as the Discussion - it should be written in a more scientific manner.”
Response 14: Thank you for your suggestion. We made modifications to our conclusion to further enrich its understanding.
Comments on the Quality of English Language: “The authors use very long sentences which causes the loss of the whole concept. Few examples - lines 293-298; 324-330; 346-350. The use of meanwhile and nonetheless in the beginning of some paragraphs in the Discussion section is inappropriate. Throughout the hole text there are a lot if word which are used inappropriate.”
Response: We appreciate your comment. Changes were made to the sentences that you pointed out, as well as the whole discussion.
Reviewer 4 Report
Comments and Suggestions for Authors
- It is suggested that the introduction be summarized
- It is unclear which RT-qPCR kit was used to identify the SARS-CoV-2 variants. Defining the variant over time would be difficult, as several variants were circulating simultaneously in some months. If this classification was based on the methodology of Padilha (2022), it must be clarified in methodology, since all the analyses depend on this classification
- The authors should include a table with the analyses of the genetic variants and RNA expression with clinical characteristics of the patients, to better understand the impact of ACE2 and TMPRSS2 on patient outcomes and not only about the virus variant. These analyses will allow them to comply with the general objective fully "this study strived to establish a relationship between ACE2 and TMPRSS2 expression profiles and genomic variations with SARS-CoV-2 variants to elucidate their possible influence on clinical outcomes of COVID-19 cases"
- Table 1 should clarify which protein is affected by each SNP, as two of the SNPs listed are from ACE2 and the other two are from TMPRSS2.
- The discussion section should include information regarding the impact of SNPs on the structure, function, and affinity of proteins.
Author Response
Comment 1: “It is suggested that the introduction be summarized”
Response 1: We are grateful for your suggestion. Agreed, changes were made in lines 35 and 69.
Comment 2: “It is unclear which RT-qPCR kit was used to identify the SARS-CoV-2 variants. Defining the variant over time would be difficult, as several variants were circulating simultaneously in some months. If this classification was based on the methodology of Padilha (2022), it must be clarified in methodology, since all the analyses depend on this classification”
Response 2: Thank you for your comment. SARS-CoV-2 variants were identified via genomic sequencing (lines 166 – 177). Although we understand your suggestion, since the first detection of SARS-CoV-2 in the samples was made using RT-qPCR, it was done in a previous study (lines 105 – 115).
Comment 3: “The authors should include a table with the analyses of the genetic variants and RNA expression with clinical characteristics of the patients, to better understand the impact of ACE2 and TMPRSS2 on patient outcomes and not only about the virus variant. These analyses will allow them to comply with the general objective fully "this study strived to establish a relationship between ACE2 and TMPRSS2 expression profiles and genomic variations with SARS-CoV-2 variants to elucidate their possible influence on clinical outcomes of COVID-19 cases"”
Response 3: Your suggestion is much appreciated. Agreed, however, the symptomatologic data we gathered from patients isn´t enough to propose a possible relation with ACE2/TMPRSS2 expression and SNPs. But you´re right in suggesting that it would be better if we had that data to support the above-mentioned objective. Given that, we changed the objective in line 102 to better fit our study.
Comment 4: “Table 1 should clarify which protein is affected by each SNP, as two of the SNPs listed are from ACE2 and the other two are from TMPRSS2.”
Response 4: We thank you for your comment. Agreed, the suggested changes were made in Table 1.
Comment 5: “The discussion section should include information regarding the impact of SNPs on the structure, function, and affinity of proteins.”
Response 5: Thank you for your comment. In the discussion we only explored the rs2285666 SNP, in the literature it has been shown that it alters ACE2 expression via mRNA splicing, however, the structural influence is yet to be explored. Still, we included in lines 340 and 341 the mRNA splicing attribute of the SNP.
Round 2
Reviewer 2 Report
Comments and Suggestions for Authors
All comments were addressed.
Reviewer 3 Report
Comments and Suggestions for Authors
Thank you for the response and the updated version of the manuscript.
Comments on the Quality of English LanguageIt is highly recommended for the manuscript to be reviewed by either a native speaker or someone with high proficiency in English.
Reviewer 4 Report
Comments and Suggestions for Authors
Requested changes made. Although the general objective was adjusted to the results presented, it is suggested that this research continue and that the clinical characteristics of the patients associated with the genetic variants and protein expression be published in the future.